# Plant defences mediate interactions between herbivory and the direct foliar uptake of atmospheric reactive nitrogen

Stuart A. Campbell [1,2] & Dena M. Vallano[2,3]

Reactive nitrogen from human sources (e.g., nitrogen dioxide, $NO_2$) is taken up by plant roots following deposition to soils, but can also be assimilated by leaves directly from the atmosphere. Leaf uptake should alter plant metabolism and overall nitrogen balance and indirectly influence plant consumers; however, these consequences remain poorly understood. Here we show that direct foliar assimilation of $NO_2$ increases levels of nitrogen-based defensive metabolites in leaves and reduces herbivore consumption and growth. These results suggest that atmospheric reactive nitrogen could have cascading negative effects on communities of herbivorous insects. We further show that herbivory induces a decrease in foliar uptake, indicating that consumers could limit the ability of vegetation to act as a sink for nitrogen pollutants (e.g., smog from mobile emissions). Our study suggests that the interactions of foliar uptake, plant defence and herbivory could have significant implications for understanding the environmental consequences of reactive nitrogen.

[1] Department of Animal & Plant Sciences, P3 Centre for Translational Plant Science, University of Sheffield, Western Bank, Sheffield S10 2TN, UK. [2] Department of Ecology & Evolutionary Biology, Cornell University, Ithaca, NY 14853, USA. [3] Region 9 Air Division, U.S. Environmental Protection Agency, San Francisco, CA 94105, USA. These authors contributed equally: Stuart A. Campbell, Dena M. Vallano. Correspondence and requests for materials should be addressed to S.A.C. (email: stuart.campbell@sheffield.ac.uk)

As a result of human activity, inputs of reactive nitrogen (N) compounds into the atmosphere have tripled in the past 50 years, with deleterious consequences for urban and natural ecosystems[1–4]. For example, nitrogen oxides ($NO_x$) have negative effects on human health[2,5] and alter important biogeochemical processes, including carbon sequestration, N cycling and global warming[6–9]. Reactive N also poses a significant and growing threat to biodiversity[10,11]. In terrestrial ecosystems, numerous studies have demonstrated that reactive N deposition to soils is causing widespread declines in plant diversity by altering soil chemistry and disrupting competitive dynamics[3,12–14]. However, the cascading effects of reactive N on higher trophic levels remain poorly studied, and there are few data on the complex pathways by which reactive N could influence consumers[15].

One mechanism by which reactive N could affect higher trophic levels is via foliar uptake. Most research has focussed on root uptake of reactive N from soil following deposition. However, $NO_x$ gases can also be taken up directly by plant leaves and used as a nutrient source[7,16]. Up to 15% of a plant's N budget can be obtained via foliar uptake, making it a significant pathway for the cycling of reactive nitrogen[17,18]. $NO_x$ enters leaves via stomatal diffusion and undergoes apoplastic disproportionation and ascorbate scavenging before downstream metabolism. Variation in these processes within the plant and among species[19] suggests that the environmental consequences of foliar uptake will be more complex than root uptake, particularly for consumers. $NO_x$ deposited onto soil is taken up by roots as $NO_3^-$ and should simply increase plant %N and may benefit herbivores, a prediction supported by empirical studies[15,20]. In contrast, foliar $NO_x$ assimilation could have divergent effects on consumers depending on its metabolic fate: N derived from $NO_x$ uptake could be stored as free amino acids and could benefit leaf-feeding herbivores, similar to the effects of root fertilisation. Conversely, if $NO_x$ was incorporated into defensive metabolites or caused shifts in defensive secondary metabolism, then consumers would suffer reduced growth and reproduction. Negative effects of foliar uptake of reactive N for herbivores have not yet been demonstrated. The only prior studies used high-concentration, short-term fumigation and showed positive effects on aphid colony growth[21,22]. However, aphids feed on phloem, which is severely N-limited and

contains few defensive compounds[23]. Thus the potential for negative impacts of foliar $NO_x$ assimilation may have been underestimated.

In this study, we test the effects of foliar $NO_2$ uptake on plant metabolism and herbivorous insects and then test the reciprocal effects of herbivory on $NO_2$ uptake, using tobacco, *Nicotiana tabacum*, and its natural leaf-feeding herbivore, the tobacco hornworm, *Manduca sexta*. We are able to accurately trace reactive N uptake in leaves and monitor how herbivory affected reactive N assimilation by using a stable isotope fumigation system with $^{15}NO_2$. Plants exposed to $NO_2$ assimilate significant quantities of atmospherically derived N, exhibit upregulation of alkaloid defensive metabolites and support lower herbivore growth. In turn, herbivore feeding causes plant-wide reductions in foliar $NO_2$ assimilation. Our study demonstrates that, in addition to disrupting plant communities through soil deposition, anthropogenic reactive N may have extended negative consequences for higher trophic levels. Our results also indicate that insect herbivores could influence the capacity of leaves to absorb reactive N and act as a sink for these atmospheric pollutants. This feedback between foliar $NO_2$ uptake and herbivory may have implications for predicting the fate of reactive N in terrestrial ecosystems.

## Results

**Effects of foliar $NO_2$ uptake on insect herbivores.** Using sealed chambers, we grew plants from the seedling stage in enriched $^{15}NO_2$ at a concentration (40 ppb) similar to current urban levels;[24] control chambers were maintained at 0 ppb. Plant roots were suspended in a hydroponics system at either 50 mM $NO_3^-$ (low-N) or 500 mM $NO_3^-$ (moderate-N) to compare responses at two ecologically realistic root N levels; shoots and roots were segregated to allow accurate partitioning of plant N sources[18]. After 4 weeks, *M. sexta* larvae were applied to half the plants in each chamber and allowed to feed. Herbivores feeding on $NO_2$-exposed plants showed a pronounced reduction in growth (a proxy for fitness in this species[25]) compared to those feeding on plants in control chambers (Fig. 1a, Supplementary Table 1), indicating that anthropogenic $NO_2$ can have potent negative effects on plant consumers. There was also a significant negative

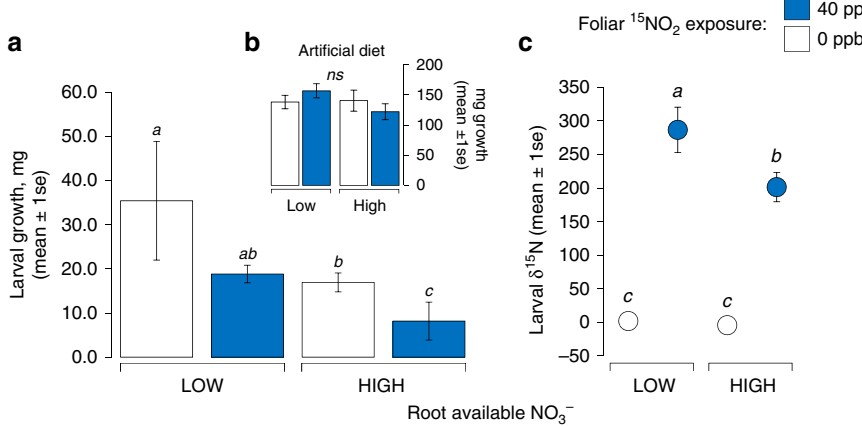

**Fig. 1** $NO_2$ indirectly reduces herbivore performance and is transferred across trophic levels. **a** Growth of *Manduca sexta* larvae on leaf tissue of *Nicotiana* plants exposed to 40 ppb $NO_2$ (blue bars) and 0 ppb $NO_2$ (white bars) under low and high concentrations of root-available $NO_3$ (10 days). **b** Inset panel shows no direct effect of $NO_2$ on larvae feeding on artificial diet in the experimental (40 ppb $NO_2$) chambers. **c** $\delta^{15}N$ values of tissue from larva feeding on plants in chambers. Within each plot, different combinations of letters denote significant differences among means (Tukey's test, $P < 0.05$), where present; ns indicates not significant. Data represent $N = 3$ chamber averages with $N = 4$ plants per chamber, and error bars are ±1 SE, with full statistical results provided in Supplementary Table 1

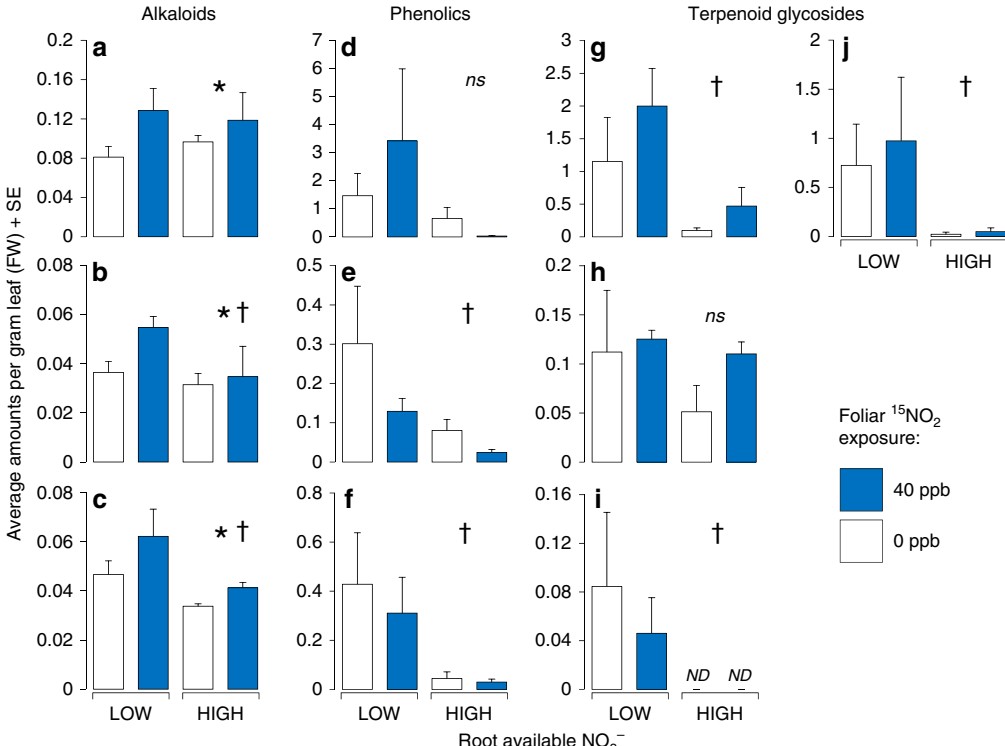

**Fig. 2** Analysis of the *Nicotiana* defensive metabolome under NO₂. Foliar metabolite profiles of *Nicotiana* plants exposed to 40 ppb NO₂ (blue bars) and 0 ppb NO₂ (white bars) under low and high concentrations of root-available NO₃, based on three classes of defence-related secondary metabolites (alkaloids, phenolics and diterpene glycosides). **a** Nicotine; **b** anatabine; **c** alkaloid 3; **d** chlorogenic acid; **e** caffeic acid 2; **f** caffeic acid 3; **g** terpenoid glycoside 1; **h** terpenoid glycoside 2; **i** terpenoid glycoside 3; **j** terpenoid glycoside 4. Alkaloid (**a–c**) and phenolic (**d–f**) quantities are given as μg gFW⁻¹; terpenoid glycosides (**g–j**) are given as peak-area gFW⁻¹. An asterisk (*) denotes a significant main or interactive effect of NO₂ treatment; a dagger (†) denotes a significant main or interactive effect of NO₃ treatment and ns indicates not significant (GLMM). Data represent $N = 3$ chamber averages with $N = 4$ plants per chamber, and error bars are +1 SE, with full statistical results provided in Supplementary Table 2

effect of higher $NO_3^-$ on larval performance, which was independent of $NO_2$ level (Supplementary Table 1). The effect of $NO_2$ was not due to toxicity from direct exposure: larvae feeding on artificial diet in the chambers showed no adverse effects (Fig. 1b), confirming that plant quality had been adversely affected by $NO_2$. Analysis of $\delta^{15}N$ values in larval tissues (excluding gut tissues and plant material) revealed that $NO_2$-derived nitrogen was used by herbivores for growth (Fig. 1c). The negative effect of $NO_2$ on herbivore performance suggests that, in contrast to the possible benefits to herbivores of soil N deposition[15], atmospheric reactive N will increase plant resistance and have important negative consequences for plant consumers when taken up by leaves.

**Effect of foliar NO₂ uptake on plant-defensive metabolites.** The genus *Nicotiana* has a well-characterised suite of defence-related secondary metabolites[26], allowing us to test whether $NO_2$ affected insects by altering the expression of defensive compounds in leaves. Targeted metabolomic analyses revealed that increased herbivore resistance under $NO_2$ was associated with significant increases in foliar alkaloids, a class of N-rich, toxic defensive compounds (Fig. 2, Supplementary Table 2). Levels of three alkaloids were all significantly higher in plants exposed to $NO_2$, including the dominant defensive compound nicotine, levels of which were on average 45% higher under $NO_2$ exposure. Total leaf alkaloid content did not differ significantly between low and high $NO_3^-$ levels, indicating that the effect of $NO_2$ uptake on defence was not based on overall N availability; instead, dissolved $NO_2$ in the apoplast may represent a mobile N pool available for transport to the site of alkaloid biosynthesis (roots). In support of this interpretation, total alkaloid content of plants under 40 ppb

$NO_2$ was positively correlated with the amount of $^{15}NO_2$-derived N, while remaining independent of total N (Supplementary Figure 1). It is possible that $NO_2$ acted as a stress or stress signal[27] causing upregulation of plant defences, but these interpretations are not supported by the lack of effect of $NO_2$ either on plant growth (Supplementary Figure 2a, Supplementary Table 3) or on two other classes of defensive metabolites (phenolics and terpenoids) (Fig. 2, Supplementary Table 2). Instead, carbon-based phenolic and terpenoid compounds were strongly reduced under higher root nitrogen ($NO_3^-$) (Fig. 2, Supplementary Table 2). This result is consistent with theories on plant defence, which predict reduced C-based defences in high-nutrient environments that allow plants to produce potent N-limited defences (e.g., alkaloids) and/or tolerate damage[28]. Lower herbivore performance on high $NO_3^-$ plants was not explained by metabolite variation, suggesting that levels of an unmeasured defence trait (e.g., proteinase inhibitors) may have been higher in the high $NO_3^-$ treatment. Our results demonstrate that plant defences will be sensitive to N deposition to soil but will also be uniquely sensitive to N derived from foliar uptake, with the potential to affect a wide range of organisms that interact with a plant's secondary metabolome, including herbivores, pollinators and microbes.

**Effect of herbivory on foliar NO₂ uptake.** The use of $^{15}N$-enriched $NO_2$ allowed us to test whether herbivore-induced changes to plant physiology would influence foliar uptake of $NO_2$ and its allocation among plant tissues. Plants exposed to herbivory contained 36% less $NO_2$-derived N (Fig. 3). Larger total amounts of $^{15}N$ were incorporated under high root $NO_3^-$, consistent with the larger plant size in this treatment (Supplementary

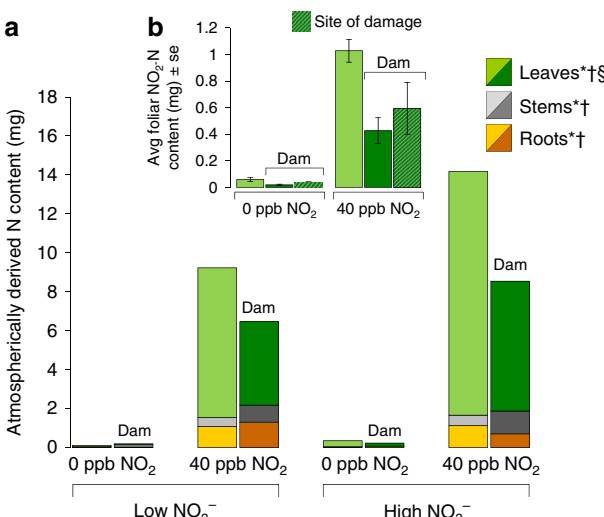

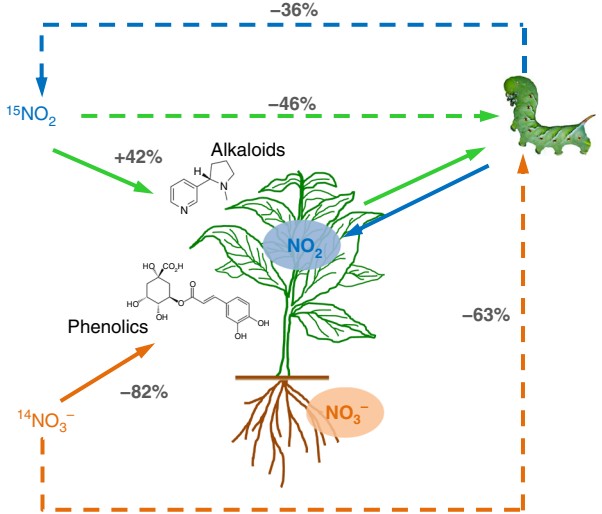

**Fig. 3** Root N availability and herbivory alter foliar uptake of $NO_2$.
**a** Distribution of $NO_2$-derived N, as measured by average of the total mg of $^{15}N$ in root (orange bars), stem (grey bars) and leaf (green bars) tissues under low (50 mM) and high (500 mM) levels of $NO_3$ in the presence and absence of herbivore damage ("Dam"). Data represent $N = 12$ plants.
**b** Foliar uptake under 40 ppb $NO_2$ in plants growing at low (50 mM) $NO_3$, following damage and induction by herbivores. Light green bars are data for control plants. Dark green bars represent data for herbivore-exposed plants ("Dam"); hatched bars are data for the site of herbivore feeding, while unhatched bars are values for systemically induced leaves that were not directly consumed. Data represent $N = 6$ plants, and error bars are ±1 SE. For each tissue type, an asterisk (*) denotes significant effect of $NO_2$ treatment, a dagger (†) denotes a significant effect of $NO_3$ treatment and a section sign (§) denotes a significant effect of herbivore exposure. Full statistical results provided in Supplementary Tables 4 and 6

**Fig. 4** Summary of feedbacks between $NO_2/NO_x$, $NO_3$, plants and herbivores. Direct effects are denoted by solid arrows and indirect interactions by dashed arrows. $NO_2$ caused significant increases in total foliar alkaloids, e.g. nicotine (solid green arrow), and a corresponding decrease in herbivore performance. In turn, feeding by herbivores caused a decrease in foliar $NO_2$ uptake. Plants in high $NO_3$ conditions decreased investment in C-based defensive metabolites, but herbivores on these plants performed relatively poorly. $NO_2$ exposure and $NO_3$ level did not strongly influence C-based and N-based defensive secondary metabolite production, respectively

even in urban environments, could represent a significant control on how readily plants assimilate and sequester reactive N pollutants.

Figure 2a, Supplementary Table 3). Lower $^{15}N$ in damaged leaf tissue likely resulted in part from differential efflux of $NO_2$ metabolites from leaves following uptake, as indicated by the distribution of $^{15}N$: herbivory decreased the proportion of the plant's total $^{15}N$ in leaves by an average of 16% and caused a corresponding increase of 93% in roots, after accounting for the net reduction in total uptake (Fig. 3a, Supplementary Table 4). The effect of herbivory on $^{15}N$ content did not appear to be due to changes in overall leaf %N (Supplementary Figure 2). The reduction in total $^{15}N$ across all tissues suggested that herbivory also directly reduced foliar uptake rates as a result of induced changes in plant metabolism and/or stomatal conductance[29]. To independently test this hypothesis, we grew an additional set of *N. tabacum* plants at 50 mM $NO_3^-$, damaged half with neonate *M. sexta* (5–7% leaf area removed) and then exposed them to 40 ppb $^{15}NO_2$. Prior feeding by herbivores caused significant reductions in $^{15}N$ assimilation in both damaged and undamaged leaves, confirming that herbivory reduced plant-wide foliar uptake (Fig. 3b, Supplementary Table 6). Consistent with results of our main experiment, $NO_2$ exposure caused significant increases in overall foliar %N that were offset by reductions in damaged plants (Supplementary Figure 3, Supplementary Table 6). These results indicate that herbivory will strongly modulate the role of vegetation as a sink for atmospheric $NO_x$ compounds in at least two ways: first, by altering the physiological controls on $NO_2$ uptake (i.e., metabolism and/or stomatal behaviour), and second, by reducing plant size through consumption (physical leaf area). The systemic reduction in $^{15}N$ assimilation under relatively low damage (<10%) suggests that herbivores, which are ubiquitous

## Discussion

We have identified a feedback between atmospheric reactive N, plants and consumer fitness that has implications for our understanding of plant responses to global change (Fig. 4). First, $NO_2$ caused increases in N-based plant defences and reductions in herbivore performance, suggesting that increasing levels of $NO_2$ could have profound consequences for consumers, particularly in urban ecosystems. In turn, herbivore-induced reductions in foliar uptake may impose a limit on the effects of $NO_2$ on defensive chemistry. However, the stability of this feedback remains unclear: while low herbivory caused pronounced reductions in $NO_2$ uptake, the dose–response relationship between $NO_2$ and defence remains unknown. In one scenario, plants in high $NO_x$ environments would exhibit chronically elevated defences, with negative consequences for consumers, while the presence of even diminished herbivore populations could reduce the beneficial sequestration of $NO_x$ and foliar cycling of reactive N. Models of ecosystem functioning under global change rarely consider interactions with herbivores. However, in a recent study of $CO_2$ enrichment, herbivory significantly reduced plant carbon sequestration[30], which, together with our results, indicates a need to better understand how consumers will influence ecosystem and plant-level responses to different atmospheric changes.

Our study also suggests that $NO_x$ emissions will interact with other global change drivers with synergistic negative effects on consumers. For example, elevated carbon-to-nitrogen (C:N) ratios in plants growing under rising $CO_2$ are predicted to challenge herbivores to increase consumption rates in order to maintain a balance of limiting nitrogen[31,32]. However, herbivory

will also depend critically on interactions among other anthropogenic changes, including the effects of $NO_x$ and $CO_2$ on plant defences, and effects of N deposition on plant %N and responses to $CO_2$[33–35]. Moreover, in our experiments herbivore performance was not related to the effects of $NO_x$ on leaf C:N or %N (Supplementary Figure 2), highlighting the importance of understanding how reactive N, $CO_2$ and other anthropogenic factors jointly influence defensive metabolites, in addition to specific nutritive indicators (e.g., soluble protein). Our data suggest that there will be significant consequences of these interactions for plant-feeding insects, including agricultural pests and rare species already vulnerable to the effects of soil N deposition on host plant abundance[11]. Recent demonstration of pronounced declines in arthropod diversity and abundance in regions with long histories of industrialisation[36,37] further underscores the need to understand the wide-ranging environmental impacts of atmospheric reactive N on plant and insect communities.

## Methods

**Plant and insect material.** We used a model plant–herbivore system consisting of tobacco, *N. tabacum* L., (Solanaceae) and its natural leaf feeding herbivore, the tobacco hornworm, *M. sexta* L. (Lepidoptera: Sphingidae). Tobacco was selected because it is a fast-growing herbaceous species that grows well in hydroponic systems, exhibits significant N reduction in leaf tissue[38] and has a well-characterised suite of defence-related secondary metabolites[39]. Tobacco hornworm was chosen because it is a native Solanaceae specialist that readily feeds on tobacco and rarely leaves a tobacco host plant with sufficient leaf material; experimental larvae were obtained from a large in-house colony. Seeds (*N. tabacum* x. *sandera*) were purchased from a commercial producer (Paramount Seeds, Inc., Palm City, FL, USA) and germinated and grown in perlite (Sun-Gro Horticulture, Bellevue, WA, USA) in a climate-controlled growth chamber (EGC, Chagrin Falls, OH, USA) at day and night temperatures of 27 and 21 °C, respectively, under moderate light (700 $\mu$mol m$^{-2}$ s$^{-1}$) and a 16-h photoperiod. Plants were watered daily to saturation for 2 weeks and fertilised with a Hoagland's solution containing $NO_3^-$ of known $\delta^{15}$N. Forty-eight 2-week-old plants were transplanted into the hydroponics–fumigation system and used for the experiment. The remaining 12 plants were harvested and measured for biomass, leaf area and isotopic composition to provide baseline data for the experimental plants.

**Hydroponics–fumigation system and experimental design.** We used a factorial design in which we manipulated foliar exposure to atmospheric reactive N (simulating pristine and urban environments), together with the availability of root N (low vs. moderate), using a coupled hydroponics–fumigation system. The system consisted of 4, 50-L polyethylene nutrient tanks (120 × 58 × 16 cm$^3$) each fitted with three airtight Plexiglas fumigation chambers (36 × 25 × 43 cm$^3$) with an opaque base. The system was located in a greenhouse with day and night temperatures of 27 and 21 °C, respectively, relative humidity of 60–70% and moderate light conditions (800 ± 75 $\mu$mol m$^{-2}$ s$^{-1}$) using natural and supplemental metal halide lighting (400 MH Econo Cool Grow Light, Sunlight Supply, Vancouver, WA, USA) under a 16-h photoperiod. Roots were suspended in nutrient solution (20 °C) in the hydroponics tank via small holes in the chamber base, and shoots were enclosed in fumigation chambers (ambient temperature). Plants were stabilised and fitted with modelling clay (Loctite, Henkel Consumer Adhesives, Avon, OH, USA) at the root–shoot junction at the base of each chamber to ensure an impermeable seal between the fumigation and nutrient solution system components.

Fumigation chambers were supplied with activated charcoal-filtered, ambient air using a reciprocating air compressor (Model C403L, Gardner Denver, Quincy, IL, USA) at a flow rate of 15 L min$^{-1}$. Half the chambers served as controls, and half the chambers received enriched $^{15}NO_2$ from a compressed tank (1% $NO_2$: 99% $N_2$, Scott Marrin, Inc., Riverside, CA, USA), which was diluted into the filtered air of randomly selected chambers using high-precision rotometers and mass flow controllers (Models 03216–34 and 32044–00, Cole-Parmer, Vernon Hills, IL, USA) at a fixed partial pressure (40 ppb). $NO_2$ was selected as the atmospheric N source because it is a common atmospheric reactive N compound. The two fumigation treatments were selected to simulate pristine (0 ppb) and realistic 1-h urban (40 ppb) atmospheric $NO_2$ concentrations[24,40–44]. The $\delta^{15}$N of the enriched $^{15}NO_2$ was 1720 ± 17‰, providing a large signal separation from the $NO_3^-$ source. $NO_2$ and nitric oxide (NO) concentrations were monitored using a chemiluminescence NO-$NO_2$-$NO_x$ analyser (TECO Model 42, Thermo Environmental Instruments, Inc., Franklin, MA, USA). Exhaust air from each chamber was filtered using activated charcoal and exited the system through an output line extending outside the greenhouse. Temperature and relative humidity within the chambers were monitored using humidity and temperature probes (Model HMP45A, Vaisala, Inc., Boulder, Colorado, USA) connected to a datalogger (Model CR10x, Campbell Scientific, Inc., Logan, Utah, USA).

Two $NO_3$ regimes, simulating low (50 $\mu$M) and moderate (500 $\mu$M) N availability, were applied to roots using nutrient solutions with fixed concentrations of $NO_3^-$ as the sole root N source in a modified quarter-strength Hoagland's solution[45]. The $\delta^{15}$N value of the $NO_3^-$ nutrient solutions was −0.56 ± 0.1‰. The nutrient solutions were vigorously aerated at all times to provide adequate oxygenation and ensure complete mixing and pH was maintained at 5.8–6.2 by addition of either KOH or $H_2SO_4$. A 130-L reservoir of stock solution was used for each experimental N treatment. Nutrient solutions were replaced weekly to minimise microbial activity and prevent N depletion, and $NO_3^-$ concentrations were measured weekly using an auto-analyser (Astoria Pacific, Inc., Clackamas, OR, USA).

Each of the four treatment combinations ([1] 0 ppb $NO_2$ and 50 $\mu$M $NO_3^-$, [2] 40 ppb $NO_2$ and 50 $\mu$M $NO_3^-$, [3] 0 ppb $NO_2$ and 500 $\mu$M $NO_3^-$, and [4] 40 ppb $NO_2$ and 500 $\mu$M $NO_3^-$) was replicated in three chambers ($N = 12$) with four plants/chamber. After 4 weeks of growth under these treatment conditions, 3 neonate (freshly hatched) *M. sexta* larvae were placed on two plants in each chamber (6 larvae per chamber). We began the herbivory treatments at 4 weeks to maximise the period of NO2 exposure while ensuring that plants had sufficient space within the chambers and did not initiate reproduction (which can alter defence trait expression). To minimise herbivore movement and ensure that feeding was confined to the selected plants, we used the pair of plants at a randomly selected end of the rectangular chamber and visually inspected plants several times per day. Two small cups filled with a wheat-germ diet[46] and a single neonate larva were also placed in each chamber to test for effects of unforeseen environmental variation and any direct toxic effects of $NO_2$ exposure. Larvae on both plants and diet were allowed to feed for 10 days and were then removed, allowed to purge their gut contents and peritrophic membranes (gut lining) and weighed as a proxy for performance/fitness[25]. Individual larvae were then killed by freezing and freeze-dried for subsequent stable isotope analysis; allowing natural purging of the gut lining allowed us to infer insect $^{15}$N assimilation without plant contaminants (see next). Plants were harvested for analysis following removal of the herbivores.

**Morphological and stable isotope analyses.** Whole-plant samples were separated into leaf, shoot and root tissue. Shoot length and leaf number were measured for each individual and leaf area was estimated using a leaf area meter (LI-3100 Area Meter, LI-COR, Inc., Lincoln, NE, USA). Plant samples were then dried, weighed and sub-samples analysed for tissue N and C content and $\delta^{15}$N and $\delta^{13}$C. Previously separated tissue samples were rinsed with deionised water to remove any $NO_2$ deposited to the leaf surface and dried for 3 days at 55 °C in a drying oven. Dried plant and insect tissue samples were weighed, ground to a fine powder with a mortar and pestle and sub-samples of 2.55–3.15 mg were weighed using a microbalance (Model 4504MP8; Sartorius Corp. Edgewood, NY, USA). Tissue N and C contents were measured using a CHN elemental analyser (Model Carlo Erba NC2500; Thermo Finnigan, San Jose, CA, USA). Tissue $\delta^{15}$N and $\delta^{13}$C were measured using a continuous flow isotope ratio mass spectrometer (Model Delta Plus; Thermo Finnigan, San Jose, CA, USA). All analyses were conducted at the Cornell Stable Isotope Laboratory.

**Calculation of N source partitioning.** Partitioning of plant N among sources (gaseous $NO_2$ and nutrient solution $NO_3^-$) was calculated using a two-ended linear mixing model[47] and published fractionation factors for root $NH_4^+/NO_3^-$ assimilation[48]. Because we used an artificially high enrichment of $^{15}$N in the $NO_2$ fumigation, fractionation events associated with foliar uptake were likely not detectable (i.e., the signal separation generated by the tracer greatly exceeded natural fractionations). Using this model, we estimated the total amount of $NO_2$ incorporated via direct foliar uptake during the fumigation period for leaf, stem and root tissues in each experiment. In testing for larval uptake of $NO_2$-derived N, we used the $\delta^{15}$N values directly due to unknown fractionation factors associated with sequential larval instars[49] and since larvae differed in their developmental stage due to the treatments.

**Plant chemistry.** We estimated the levels of defence-related secondary metabolites in control (undamaged) plants using protocols developed for the major classes of tobacco defensive compounds, specifically pyridine alkaloids (e.g., nicotine, anatabine) phenolic compounds (caffeic acids and flavonoids) and terpenoid glycosides[50]. Considerable prior research has established these compounds as defence-related metabolites in tobacco, *N. tabacum*, and other *Nicotiana* species, e.g. refs. [39,51], and defence expression in *Nicotiana* can be sensitive to root N availability[52]. We used control plants to measure the effects of $NO_2$ and $NO_3$ treatments on plant secondary metabolism; we could not assess herbivore-induced plant responses in leaf chemistry, since herbivory was variable among treatments (see Results), and variation in herbivory leads to differentially induced metabolite expression. Fresh tissue from a fully expanded leaf was excised (avoiding the midvein), weighed (ca. 100 mg), flash frozen, ground to a fine powder in liquid $N_2$ using a pestle and stored at −80 °C. We homogenised samples using a FastPrep® tissue homogeniser (MP Biomedicals® LLC, Santa Ana, CA, USA) at 6 m s$^{-1}$ for 90 s using 0.9 g of grinding beads (Zirconia/Silica 2.3 mm, Biospec® Products Inc., Bartlesville, OK, USA) with 1 mL of ice-cold 40% methanol and 0.5% acetic acid solvent. Samples were centrifuged and a 15-$\mu$L aliquot of supernatant was analysed by high-performance liquid chromatography (HPLC) using an Agilent® 1100 series

HPLC-DAD equipped with a Gemini C18 reverse-phase column (3 μm, 150 × 4.6 mm$^2$, Phenomenex Inc., Torrance, CA, USA) and a standard method[50]. Alkaloid, phenolic and diterpene glycoside analytes with identifiable ultraviolet spectra were selected and initially quantitated by peak area. Individual compound identification of nicotine, anatabine and chlorogenic acid was based on comparison with authentic standards, and peak areas were converted to μg gFW$^{-1}$ using standard curves. A third, unidentified pyridine alkaloid (Alkaloid 2) and two unidentified caffeic acid derivatives (Caffeic acid 2 and Caffeic acid 3) were converted to mass equivalents of nicotine and chlorogenic acid, respectively. All analyte quantities were normalised by the fresh sample masses prior to statistical analysis.

**Test for herbivore-induced changes to foliar uptake**. We observed indications of reduced incorporation of $^{15}$N under herbivory in our main experiment, suggesting the potential for herbivores to induce changes to foliar $NO_2$ assimilation. To test this hypothesis and distinguish effects of herbivore-induced plant responses on uptake from effects of induced responses on N allocation (see Results), we conducted a second experiment: We grew a new group of 24 *N. tabacum* plants in the hydroponics–fumigation system at 50 μM $NO_3^-$ and 0 ppb $^{15}NO_2$ for 2 weeks, at which point we applied four neonate *M. sexta* on each of two plants per chamber, as in the main experiment. Larvae fed for 2 days, at which point damage levels were ca. 5–7%, which is sufficient to cause an induced response in *Nicotiana*[26]. Half the chambers then received $^{15}NO_2$ fumigation at 40 ppb for a subsequent 5 days, and half served as controls (0 ppb). During this period, plants were inspected and 5–6 additional larvae were added to a few plants to maintain similar total damage levels (ca. 10% on each of four leaves) across treatments. Leaf tissue was harvested from control plants and from the damaged and undamaged leaves of the herbivore-treated plants and analysed for incorporation of $^{15}$N from $NO_2$ uptake in the four treatments (0 ppb $NO_2$/Control; 0 ppb $NO_2$/Herbivory; 40 ppb $NO_2$/Control; 40 ppb $NO_2$/Herbivory). Analysis of the damaged (locally induced) and undamaged leaves on damaged plants allowed us to test whether herbivory had caused a systemic (plant-wide) induced change in foliar $NO_2$ uptake.

**Statistical analysis**. Data were confirmed for normality and analysed using standard general linear models in a maximum likelihood framework (JMP® v.13). Larval performance data, larval $^{15}$N uptake and plant metabolite data were analysed by two factor models, with nutrient solution ($NO_3^-$ level) and $NO_2$ fumigation level as fixed factors and their interaction. For consistency, the data for larval performance on diet were analysed in the same way, though there was no predicted effect of $NO_3^-$. Morphological data were analysed by three factor models with $NO_3$, $NO_2$ and herbivory as fixed factors and their interactions. Data were averaged within individual chambers; analyses with plant average as the observational unit did not qualitatively change the results, indicating that our experiment had sufficient statistical power for the observed effect sizes for these response variables. Plant $^{15}$N uptake, %N and C:N data were analysed by three-factor models that included $NO_3^-$, $NO_2$ and herbivory levels. We used plant averages for these analyses to compensate for the reduced power available in higher-order models; however, plant-level and chamber-level averages again gave similar results. We analysed foliar uptake data using separate models for (a) damaged leaves, (b) undamaged leaves and (c) pooled leaf data from plants exposed to herbivores, in order to verify that plant responses were consistent at the local and systemic (plant-wide) scale. Means and standard errors for uptake rates for all treatments and tissue types are provided in Supplementary Table 7. Finally, the effect of $NO_2$ exposure on secondary metabolites prompted us to determine whether there was a quantitative relationship between e.g. alkaloid content and the amount of $NO_2$-derived N taken up by leaves. For this, we used plants in the 40 ppb $NO_2$ treatment and tested for correlations (restricted maximum likelihood estimation) between total amounts of each metabolite class in each plant and the amount of $^{15}NO_2$-derived N and total percentage of nitrogen in leaves.

## Data availability
The data that support the findings of this study are available from the corresponding author upon reasonable request.

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

## Acknowledgements

We thank A. Kessler (Cornell University) for generous use of the HPLC instrument and advice; K.E. Sparks, J.P. Sparks and A. Kasson (Cornell Stable Isotope Laboratory) and R. Halitschke (Cornell University; Max Planck Institute for Chemical Ecology) for technical assistance and advice; M. del Campo for assistance with *Manduca sexta*; and P. Cooper for assistance with greenhouse management. This study was supported by a Biogeochemistry and Environmental Biocomplexity Initiative Small Grant awarded to S.A.C. and D.M.V. through a National Science Foundation (U.S.A.) Integrative Graduate Education and Research Traineeship (NSF-IGERT DGE-0221658); S.A.C. was supported by a Natural Sciences and Engineering Research Council (NSERC Canada) PGS-D fellowship and an Independent Research Fellowship from the University of Sheffield P3 Centre for Translational Plant Science (BB/IAA/Sheffield/15). M. Stastny, A. Kessler and J. Sparks provided feedback on the project and/or drafts of this manuscript.

## Author contributions

S.A.C. and D.M.V. conceived and designed the study, collected and analysed the data and wrote the paper.

## Additional information

**Competing interests:** The authors declare no competing interests.

