## [Peer Review File · Nature Communications]

Reviewers' comments:

Reviewer #1 (Remarks to the Author):

Campbell and Vallano provide details on an elegant experiment to test the impacts of N-based air pollution on herbivory and plant response. This is an important step forward in understanding potential mechanisms for N deposition impacts on natural and managed systems. While such links have been posited in the literature, this study is the first that I am aware of that provides convincing data addressing the important relationship. My comments are limited to relatively minor suggests for improvements related to presentation and clarity.

Use of larval growth as a measure of fitness is not consistent with the definition of fitness in the evolutionary sense. While a disclaimer (L82) indicates the intended use as a fitness proxy, this use is still in my mind rather problematic (e.g., Fig. 1: "NO₂ indirectly reduces the fitness of herbivores..."). I do not think it would weaken the claims of the paper to simply refer to the measured data as larval growth.

Extended data fig 1a - Units are missing from the y-axis legend and figure legend.

Extended data fig 2 - I believe that I eventually figured out the meaning of the colors based on the Figure 3 legend. However, a clear description in this figure legend would be helpful.

Extended data Table 1 - As I understand it "Growth on diet" refers to larvae grown on the artificial diet, with or without NO₂ exposure. It is not clear to me how NO₃ and a NO₂ x NO₃ interaction are part of the "Growth on diet model". Perhaps I interpreted the methods incorrectly, but I could not find any information about a NO₃ treatment for the artificial diet study.

Extended data Table 2 - The figure legend appears to have an incorrect reference to figure 1. The "labels" column appears to refer to figure 2.

Reviewer #2 (Remarks to the Author):

The manuscript reports on experimental work on the uptake of reactive nitrogen oxides by shoots of *Nicotiana* plants directly from the atmosphere and its consequences for the specialist leaf-chewing herbivorous larvae of the tobacco hornworm *Manduca sexta*. The findings are novel since the negative effects of foliar uptake of reactive atmospheric NO_x for an insect herbivore has not previously been demonstrated. The study also addresses the effect of herbivory on NO_x uptake and reveals a significant decrease.

The methods and statistics are elaborate and sound and the results solid and important for a wider audience because of their potential ecological impacts. I do have some reservation about the too broad generalization to insect-plant interactions since many of such interactions do not rely on N-containing defensive secondary metabolites.

The regulation of the underlying biosynthetic mechanisms operating in the plant remain unknown. In particular for the possible incorporation of NO_x into defensive alkaloids the authors missed an obvious opportunity which would be a relevant complementation of the present data.

The title of the manuscript does not adequately reflect the actual findings. The feedback mechanism is not clearly explained in the manuscript; one would expect that it becomes evident from Figure 4, which presents a useful pictorial summary of the findings but could be improved for clarity (see below). Consider to rephrase the title to become more straightforward. Rather than a feedback, an 'interactive effect' would in my view be a more appropriate term. 'Feedback' without either of the qualifiers 'positive' or 'negative' is not informative.

- I. 62: 'should simply increase plant %N balance': suggest to rephrase: 'is expected to increase plant nitrogen content'.
- I. 77: Is 0 ppb NO_x a concentration that occurs in pristine outside air? Provide reference(s).
- I. 80: After 4 weeks: explain why this particular time point was chosen?
- I. 82: Either provide a reference that demonstrates that growth in early larval instars is a proxy for fitness of *Manduca sexta*. If no such reference exists, rephrase to what was actually quantified: 'larval growth performance'.
- I. 83: Figure 1: How can the occurrence of NO₂-derived N in plants not exposed to NO₂ be explained?
- I. 86-87: the body weight of larvae fed artificial diet is much higher than that of larvae fed on *Nicotiana* plants. Specify if the experiment was started with the same larval instars? In addition, it should be explained how the low and high nitrate levels in the hydroponic solution should be interpreted in the artificial diet experiment?
- I. 88-89: The observation that the ¹⁵N-isotope was found in the larval body does not prove that it is used for growth; it rather shows that it has been absorbed from the food into the haemolymph and may serve other functions than tissue growth.
- I. 102-103: It is missed opportunity that the ¹⁵N incorporation into alkaloids has not been quantified. Without these data, the statement is speculative and the suggested transport of nitrogen derived from NO₂ from shoot to root to be incorporated into alkaloids is a long haul.
- I. 104: Should read: '...upregulation of plant defences...'.
I. 119: the reduced ¹⁵N-uptake by local leaves can only be correctly interpreted when the amount of leaf material removed by herbivory has been quantified. In I. 139 '< 10%' is stated; it is more clear to mention this here and be compared with the reduction in foliar ¹⁵N uptake.
- I. 129-130: '(5-7% damage)': this means 5–7 % of leaf surface removed?
- I. 144-146: The statement made here is too general, since it is valid only for nitrogen-based defences.
- I. 343: 'larval fitness' should read 'larval performance'.

Figure 3, Legend: it should be indicated what light green represents. Herbivore exposure (H) is used as identical to damage (Dam); use either one consistently.

Figure 4: The lay-out of the figure can be improved in my view.

The green dashed arrow should start at the drawing of the alkaloid molecule, since the effect is not due to ¹⁵NO₂ but due to higher alkaloid levels in the plant.

The caterpillar is better positioned on a leaf, close to a picture of a damaged leaf edge in e.g. an inset, from which the blue dashed arrow then departs to point at ¹⁵NO₂.

The orange dashed line should depart from a leaf and then point to the caterpillar; nitrate is first assimilated into primary and secondary plant compounds that in turn affect caterpillar growth rate. The solid orange arrow pointing from nitrate to phenolics should be replaced by a dashed arrow since this effect surely is indirect.

The blue dashed line should point to the one-headed green arrow pointing to the alkaloid structural formula, since this indicates the uptake of NO₂ into the plant and incorporation into alkaloids.

Reviewer #1 (Remarks to the Author):

Campbell and Vallano provide details on an elegant experiment to test the impacts of N-based air pollution on herbivory and plant response. This is an important step forward in understanding potential mechanisms for N deposition impacts on natural and managed systems. While such links have been posited in the literature, this study is the first that I am aware of that provides convincing data addressing the important relationship. My comments are limited to relatively minor suggests for improvements related to presentation and clarity.

REPLY: Many thanks!

Use of larval growth as a measure of fitness is not consistent with the definition of fitness in the evolutionary sense. While a disclaimer (L82) indicates the intended use as a fitness proxy, this use is still in my mind rather problematic (e.g., Fig. 1: "NO₂ indirectly reduces the fitness of herbivores..."). I do not think it would weaken the claims of the paper to simply refer to the measured data as larval growth.

REPLY: Thanks for this comment. We have replaced "fitness" with "growth" or "performance" (e.g., in the caption of Fig 1) to minimise any confusion as to what trait we were measuring. The relationship between larval growth and fitness (survival and fecundity) is well-documented both generally (Benrey and Denno 1997, Honek 1993, Kingsolver and Huey 2008) and for this species in particular (Diamond and Kingsolver 2010), and to support the use of growth as a fitness proxy we have added the latter citation (addressing a related comment by Review #2).

Extended data fig 1a - Units are missing from the y-axis legend and figure legend.

REPLY: Many thanks for pointing this out – units have been added and the legend clarified.

Extended data fig 2 - I believe that I eventually figured out the meaning of the colors based on the Figure 3 legend. However, a clear description in this figure legend would be helpful.

REPLY: We have re-worked the legend in Extended Data Fig 2 for improved clarity and adjusted the legend for Figure 3 for consistency. Thanks for the suggestion.

Extended data Table 1 - As I understand it "Growth on diet" refers to larvae grown on the artificial diet, with or without NO₂ exposure. It is not clear to me how NO₃ and a NO₂ x NO₃ interaction are part of the "Growth on diet model". Perhaps I interpreted the methods incorrectly, but I could not find any information about a NO₃ treatment for the artificial diet study.

REPLY: Thanks for this comment. It is true that there was no predicted effect of the hydroponics solution on the growth of larvae in diet cups. However, in the unlikely event of any unanticipated variation among chambers (e.g., effects of shading due to the larger size of plants under high NO₃), and to facilitate comparison with larval growth on plants, it seemed sensible to keep our analysis the same for all larvae. To improve clarity, we have added the following sentence to the Methods (ll. 348-349): "For consistency, the data for larval performance on diet were analysed in the same way, though there was no predicted effect of NO₃"

Extended data Table 2 - The figure legend appears to have an incorrect reference to figure 1. The "labels" column appears to refer to figure 2.

REPLY: Corrected – thanks for pointing this out.

Reviewer #2 (Remarks to the Author):

The manuscript reports on experimental work on the uptake of reactive nitrogen oxides by shoots of *Nicotiana* plants directly from the atmosphere and its consequences for the specialist leaf-chewing herbivorous larvae of the tobacco hornworm *Manduca sexta*. The findings are novel since the negative effects of foliar uptake of reactive atmospheric NO_x for an insect herbivore has not previously been demonstrated. The study also addresses the effect of herbivory on NO_x uptake and reveals a significant decrease.

The methods and statistics are elaborate and sound and the results solid and important for a wider audience because of their potential ecological impacts. I do have some reservation about the too broad generalization to insect-plant interactions since many of such interactions do not rely on N-containing defensive secondary metabolites.

The regulation of the underlying biosynthetic mechanisms operating in the plant remain unknown. In particular for the possible incorporation of NO_x into defensive alkaloids the authors missed an obvious opportunity which would be a relevant complementation of the present data.

REPLY: We thank the reviewer for the positive assessment, and for the numerous helpful suggestions and comments. We have generally followed most of the suggestions for avoiding over-generalization (see specific replies below), and to clarify our presentation.

The title of the manuscript does not adequately reflect the actual findings. The feedback mechanism is not clearly explained in the manuscript; one would expect that it becomes evident from Figure 4, which presents a useful pictorial summary of the findings but could be improved for clarity (see below). Consider to rephrase the title to become more straightforward. Rather than a feedback, an

'interactive effect' would in my view be a more appropriate term. 'Feedback' without either of the qualifiers 'positive' or 'negative' is not informative.

REPLY: Thanks for this suggestion. It's possible that the word feedback may require greater explanation that would be suitable for a title. "Interactive effect" would also seem to require explanation of what the effect was, but as a compromise we have amended the title to "interactions between".

I. 62: 'should simply increase plant %N balance': suggest to rephrase: 'is expected to increase plant nitrogen content'.

REPLY: Changed to the more straightforward "increase plant %N" for brevity and precision.

I. 77: Is 0 ppb NO_x a concentration that occurs in pristine outside air? Provide reference(s).

REPLY: Thanks for the suggestion. Indeed, values close to 0 are observed in pristine environments, and citations for this have now been added to the Methods at I. 239 (Lee et al. 1997, Pike et al. 2010, see also Liu et al. 2018). Values less than 5ppb are also found throughout the U.S. (see <https://www.epa.gov/air-trends/air-quality-design-values>).

I. 80: After 4 weeks: explain why this particular time point was chosen?

REPLY: To clarify and better justify this approach, we have added the following to the Methods section: "We began the herbivory treatments at 4 weeks to maximise the period of NO₂ exposure while ensuring that plants had sufficient space within the chambers and did not initiate reproduction (which can alter defence trait expression)." (II. 262-264).

I. 82: Either provide a reference that demonstrates that growth in early larval instars is a proxy for fitness of *Manduca sexta*. If no such reference exists, rephrase to what was actually quantified: 'larval growth performance'.

REPLY: See comment to Reviewer 1 above; references have been added.

I. 83: Figure 1: How can the occurrence of NO₂-derived N in plants not exposed to NO₂ be explained?

REPLY: As is typical of this experimental set-up, these trace amounts of ¹⁵N are likely the result of natural incorporation from the hydroponics medium. We provided the isotopic signature of the medium in the Methods section so that it is clear that the natural ¹⁵N abundance is extremely low.

I. 86-87: the body weight of larvae fed artificial diet is much higher than that of larvae fed on *Nicotiana* plants. Specify if the experiment was started with the same larval instars? In addition, it should be explained how the low and high nitrate levels in the hydroponic solution should be interpreted in the artificial diet experiment?

REPLY: Both sets of larvae were neonates (fresh hatchlings) (see I. 261 and I.267) and we have now clarified (I. 269) that larvae on diet fed for the same period of time. It is typical for larvae reared on this diet to grow faster than on natural plant material, particularly plants with growth-reducing or toxic metabolites such as tobacco.

The performance on the diet was only meant to inform interpretation of the NO₂ effect, i.e., to let us rule out any directly toxic effects of NO₂ exposure. It would be difficult to use the diet experiment results to infer anything about the effect of NO₃⁻ in this case. The added comment in the methods (see reply to reviewer 1, above), will hopefully clarify this issue.

I. 88-89: The observation that the ¹⁵N-isotope was found in the larval body does not prove that it is used for growth; it rather shows that it has been absorbed from the food into the haemolymph and may serve other functions than tissue growth.

REPLY: Thanks for this comment. While a small amount of ¹⁵N might be expected in the haemolymph as a result of e.g., passive diffusion, the magnitude of the signal is far too high for the N to be restricted to haemolymph, meaning that it must be found in the other larval tissues. We are fairly circumspect in our interpretation of this finding, and thus haven't altered the text in this case.

I. 102-103: It is missed opportunity that the ¹⁵N incorporation into alkaloids has not been quantified. Without these data, the statement is speculative and the suggested transport of nitrogen derived from NO₂ from shoot to root to be incorporated into alkaloids is a long haul.

REPLY: This is a good idea, and in fact we did consider compound-specific isotope analysis. However, we lacked a set of samples with which to pursue this question, which will await a follow-up study. We do believe that our interpretation of the export of NO₂-derived N to roots is sound, and supported by the distribution of the ¹⁵N in the different plant tissues (discussed at II.123-125). In addition, there is a significant, positive correlation between NO₂ uptake and alkaloid content under the 40ppb treatment, which also supports our interpretation; this isn't due to overall nitrogen levels, since there is no effect of total nitrogen on alkaloids. We have added a figure showing these relationships to our Extended Data (now Extended Data Fig 1) to better support our conclusions.

This figure is referenced in the text with the following brief sentence (II. 103-105):

“In support of this interpretation, total alkaloid content of plants under 40ppb NO₂ was positively correlated with the amount of NO₂-derived N uptake, while remaining independent of total N (Extended Data Fig 1).”;

and we have added this section to the description of the statistics in the Methods (II. 361-365):

“Finally, the effect of NO₂ exposure on secondary metabolites prompted us to determine whether there was a quantitative relationship between e.g., alkaloid content and the amount of NO₂-derived N taken up by leaves. For this, we used plants in the 40ppb NO₂ treatment and tested for correlations (REML estimation) between total amounts of each metabolite class in each plant and the amount of ¹⁵NO₂-derived N, and total percent nitrogen in leaves.”

Many thanks for this comment, since we feel this information is a simple, but valuable addition to the supplementary material.

I. 104: Should read: ‘...upregulation of plant defences...’.

REPLY: Changed.

I. 119: the reduced ¹⁵N-uptake by local leaves can only be correctly interpreted when the amount of leaf material removed by herbivory has been quantified. In I. 139 ‘< 10%’ is stated; it is more clear to mention this here and be compared with the reduction in foliar ¹⁵N uptake.

REPLY: We agree entirely, and this was the motivation for the second experiment. However, this sentence introduces the general concept of herbivore induced reduction in NO₂ uptake, followed by interpretation of the main experiment; these results then motivate the experiment in which we tightly controlled the amount of damage (the 5-7%), introduced at I.129.

I. 129-130: '(5-7% damage)': this means 5–7 % of leaf surface removed?

REPLY: Yes; we adjusted our sentence for clarity, thanks for the suggestion.

I. 144-146: The statement made here is too general, since it is valid only for nitrogen-based defences.

REPLY: Sentence has been amended to “N-based defences”.

I. 343: 'larval fitness' should read 'larval performance'.

REPLY: Sentence has been amended as suggested.

Figure 3, Legend: it should be indicated what light green represents. Herbivore exposure (H) is used as identical to damage (Dam); use either one consistently.

REPLY: In this figure, the “Dam” denotes the experimental treatment, while the symbols next to the tissue represent a visual summary of the statistical results for each tissue. We see how the use of the double-struck “H” might make it unclear however and have changed this symbol to “§” for clarity.

Figure 4: The lay-out of the figure can be improved in my view.

REPLY: Thanks very much for the suggestions on the figure.

The green dashed arrow should start at the drawing of the alkaloid molecule, since the effect is not due to $^{15}\text{NO}_2$ but due to higher alkaloid levels in the plant.

REPLY: The dashed arrows summarise the net indirect effect of e.g., NO_2 on larvae (the indirect interactions). Here, the green dashed arrow shows the indirect effect of the NO_2 on larvae, as mediated by the direct effect of NO_2 on leaf alkaloids (solid green). We have changed the wording in the caption to “net indirect effect” to make this clearer. We have also slightly re-arranged the arrows to improve clarity of the direct effects.

The caterpillar is better positioned on a leaf, close to a picture of a damaged leaf edge in e.g. an inset, from which the blue dashed arrow then departs to point at $^{15}\text{NO}_2$.

REPLY: We attempted to add an inset as suggested but don't believe this adds clarity or information to the figure – in fact it tended to make the figure look crowded and less clear. However, it's true that the direct effect of herbivores on leaf uptake was less obvious, and we have added a solid blue arrow to denote the effect of the herbivory on leaf uptake – thanks for bringing that to our attention.

The orange dashed line should depart from a leaf and then point to the caterpillar; nitrate is first assimilated into primary and secondary plant compounds that in turn affect caterpillar growth rate. The solid orange arrow pointing from nitrate to phenolics should be replaced by a dashed arrow since this effect surely is indirect.

REPLY: As we note above, the dashed line is summarising the net indirect effect of NO_3 on the consumer, while the solid orange arrow is how this occurs, via the “direct” effect of the treatment on the phenolic chemistry.

The blue dashed line should point to the one-headed green arrow pointing to the alkaloid structural formula, since this indicates the uptake of NO_2 into the plant and incorporation into alkaloids.

REPLY: As noted above, these dashed lines summarise the net indirect effects, and we hope this is clearer with the slight modification to the figure legend.

- Benrey, B., and R. F. Denno. 1997. "The slow-growth-high-mortality hypothesis: A test using the cabbage butterfly." *Ecology (Washington D C)* 78:987-999.
- Diamond, Sarah E., and Joel G. Kingsolver. 2010. "Fitness consequences of host plant choice: a field experiment." *Oikos* 119 (3):542-550. doi: 10.1111/j.1600-0706.2009.17242.x.
- Honek, A. 1993. "Intraspecific variation in body size and fecundity in insects - a general relationship". *Oikos* 66 (3):483-492. doi: 10.2307/3544943.
- Kingsolver, Joel G., and Raymond B. Huey. 2008. "Size, temperature, and fitness: three rules." *Evolutionary Ecology Research* 10 (2):251-268.
- Lee, D. S., I. Kohler, E. Grobler, F. Rohrer, R. Sausen, L. GallardoKlenner, J. G. J. Olivier, F. J. Dentener, and A. F. Bouwman. 1997. "Estimations of global NOx emissions and their uncertainties." *Atmospheric Environment* 31 (12):1735-1749. doi: 10.1016/s1352-2310(96)00327-5.
- Liu, Yingjun, Roger Seco, Saewung Kim, Alex B. Guenther, Allen H. Goldstein, Frank N. Keutsch, Stephen R. Springston, Thomas B. Watson, Paulo Artaxo, Rodrigo A. F. Souza, Karena A. McKinney, and Scot T. Martin. 2018. "Isoprene photo-oxidation products quantify the effect of pollution on hydroxyl radicals over Amazonia." *Science Advances* 4 (4). doi: 10.1126/sciadv.aar2547.
- Pike, R. C., J. D. Lee, P. J. Young, G. D. Carver, X. Yang, N. Warwick, S. Moller, P. Misztal, B. Langford, D. Stewart, C. E. Reeves, C. N. Hewitt, and J. A. Pyle. 2010. "NOx and O-3 above a tropical rainforest: an analysis with a global and box model." *Atmospheric Chemistry and Physics* 10 (21):10607-10620. doi: 10.5194/acp-10-10607-2010.

REVIEWERS' COMMENTS:

Reviewer #1 (Remarks to the Author):

The authors have addressed my comments that stemmed from the original version of the manuscript.

For Figure 1, I appreciate the change from fitness to growth for this figure. However, more complete units would be helpful on the figure or figure legend (e.g., growth in what time period?).

I do not have additional comments on this revised version.

Reviewer #2 (Remarks to the Author):

In the revised version of their manuscript the authors have addressed the comments by both reviewers in a satisfactory manner by editing / adding to the text, and by providing additional information in Expanded data. The issues noted in the original version have been appropriately addressed and this has resulted in a transparent and important paper likely to receive broad interest.

Response to Referees

MS no. NCOMMS-18-08896A

REVIEWERS' COMMENTS:

Reviewer #1 (Remarks to the Author):

The authors have addressed my comments that stemmed from the original version of the manuscript.

For Figure 1, I appreciate the change from fitness to growth for this figure. However, more complete units would be helpful on the figure or figure legend (e.g., growth in what time period?).

I do not have additional comments on this revised version.

REPLY: Many thanks. This information is already in the Methods, but we have added it to the figure legend as requested.

Reviewer #2 (Remarks to the Author):

In the revised version of their manuscript the authors have addressed the comments by both reviewers in a satisfactory manner by editing / adding to the text, and by providing additional information in Expanded data. The issues noted in the original version have been appropriately addressed and this has resulted in a transparent and important paper likely to receive broad interest.

REPLY: Many thanks.